**Subject Category:**
Biology (whole organism)

ecology/environmental science/plant science

growth, *Lycoris radiata* var. *radiata*, nitrogen-use strategy, photosynthesis, storage, winter-green herb

**Author for correspondence:**
Satomi Nishitani
e-mail: nishitani.satomi@gmail.com

# Functional differences in seasonally absorbed nitrogen in a winter-green perennial herb

Satomi Nishitani[1,2], Atsushi Ishida[3], Toshie Nakamura[2] and Naoki Kachi[2]

[1]Department of Biology, Nippon Medical School, Kyonancho 1-7-1, Musashino, Tokyo 180-0023, Japan
[2]Department of Biological Sciences, Tokyo Metropolitan University, Minami-Osawa 1-1, Hachioji, Tokyo 192-0397, Japan
[3]Center for Ecological Research, Kyoto University, Hirano 2, Otsu, Shiga 520-2113, Japan

SN, 0000-0001-7723-9456; AI, 0000-0003-0620-5819; NK, 0000-0003-4187-7186

Nitrogen (N) uptake in response to its availability and effective N-use are important for determining plant fitness, as N is a major limiting resource and its availability changes both seasonally and annually. Storage organs such as bulbs are considered an adaptive trait with respect to plant N-use strategies. It is well known that N is remobilized from storage organs to satisfy the high demand for new growth that is not completely satisfied by external uptake alone. However, little is known about how this N absorbed during different seasons contributes to plant performance. By manipulating seasonal N availability in potted *Lycoris radiata* var. *radiata* (Amaryllidaceae), a winter-green perennial, we found that the N absorbed during different seasons had different effects on leaf growth and leaf N concentrations, effectively increasing the growth and survival of the plants. N absorbed during the summer (leafless period; N was thus stored in the bulb) enhanced plant growth by increasing leaf growth. Compared with the plants supplied with N during autumn (leaf flush period), the leafy plants also showed greater growth per unit leaf area despite the lower area-based photosynthetic capacity of the latter. By contrast, N absorbed during the autumn increased the leaf N concentration and thus the photosynthetic capacity, which was considered to enhance survival and growth of the plant during winter by reducing the potentially fatal risk caused by the absorption of photons under low temperature. Our findings have important implications for estimating plant responses to environmental changes. We predict that changes in seasonal

N availability impact the performance of plants, even that of perennials that have large storage organs, via an altered relative investment of N into different functions.

## 1. Introduction

Nitrogen (N) is a major limiting resource for plants in many ecosystems [1]. The amount of available N in the soil changes seasonally via plant–microbial interactions [2,3], and unpredictable year-to-year fluctuations often occur in nature. Therefore, N uptake in response to its availability and effective N-use are important for determining plant fitness. The presence of storage organs such as bulbs is an adaptive trait with respect to the N-use strategy of perennial plants. Because the seasonal availability of soil N does not always match the demand of plants, the capacity for N storage is considered essential to satisfy the high demand for new growth that is not totally satisfied by external uptake alone. In line with this idea, a large body of literature on this subject has accumulated (e.g. [4–12]). Many of these studies have quantified the relative contribution of stored N and external N to new growth and have examined the adaptive significance of these contributions in relation to habitat conditions and phenological traits.

However, to date, few studies have focused on the different functions of stored N and external N and the resulting effects on plant performance (e.g. growth, survival, reproduction). Studies involving the use of labelled N ($^{15}$N) have shown that allocation and translocation of N within a plant depend on the time of absorption [13,14], indicating that seasonally absorbed N plays different roles. Other studies have examined the effects of seasonally absorbed N on plant growth via manipulation of the levels of stored and/or external N available to plants [8,15–19]. However, we have not found any study that elucidated the different roles of seasonally absorbed N and examined how the differences affected plant performance in a wild plant species. Here, we hypothesize that N absorbed during different seasons plays different roles benefiting plant growth and survival. The present study examined this idea via experiments that manipulated the seasonal availability of N for the winter-green perennial herb *Lycoris radiata* var. *radiata* (Amaryllidaceae), which produces a relatively large bulb. We focused especially on the roles of N with respect to leaf production.

Plants that retain their leaves in winter when the majority of the surrounding plants shed their leaves have advantages with respect to light capture for photosynthesis. However, to take full advantage of this phenology, those plants need to have a photosynthetic capacity great enough to use the plethora of photons that they absorb. Because the activities of photosynthetic enzymes decrease as growth temperature decreases, winter leaves require relatively high concentrations of enzymes to maintain their photosynthetic ability; thus, leaves are more N-demanding per unit area in winter than leaves during other seasons. In several species, leaf N concentration has been shown to increase during winter or in response to experimental exposure to low temperatures [7,20–22]. By contrast, during warmer seasons, investment of limited N resources to leaf area growth (instead of enhancement of leaf N concentration) may be beneficial and lead to greater whole-leaf photosynthetic production [22–24]. Many winter-green and evergreen plants start unfolding new leaves during warm periods when the N demand per unit leaf area is relatively low. In addition, because winter-green plants have no leaves at the beginning of the growing season, the rapid development of foliage is essential for biomass production throughout the entire annual growth period [4]. It is therefore highly probable that winter-green plants invest seasonally absorbed N into different roles with respect to leaf production in accordance with their phenology and seasonal changes in growth temperature and light availability, which would enhance their growth and survival.

*Lycoris radiata* var. *radiata* is usually found in open sites with human activities, such as residential lands, riverbanks and edges of paddy fields, in Japan. This species is also cultivated as a garden plant. The plants have green leaves from autumn to spring, with maximum photosynthetic capacity during winter, and their roots are maintained and absorb N even during the leafless summer [25]. To investigate the functional differences in seasonally absorbed N, we designed experiments such that each plant could access soil N during only one or two of three fertilizer supply periods: summer (leafless period), autumn (period of leaf flush) and winter (period of active growth with fully developed foliage). We examined the effects of N absorbed during each period on the leaf growth and N concentration of the leaves and assessed the resulting plant performance via measurements of plant growth and photosynthetic ability. We expect that the N absorbed in summer enhances leaf growth, contributing to improved growth during the relatively warm season (autumn to early winter), while the N absorbed in autumn enhances the leaf N concentration in mid-winter, leading to a high photosynthetic capacity. The N absorbed in winter would not be invested into leaves because the environmental temperature increases towards spring; instead, this N would be stored in the bulb and affect the plant performance during the next growing season. To

our knowledge, this is the first report showing that a shift in the functions of N along with growth phenology effectively enhances the growth and survival of a wild plant species.

# 2. Material and methods

## 2.1. Plant materials

To prepare potted plants for the fertilizer supply experiment, we obtained bulbs (free of leaves and roots) of *L. radiata* var. *radiata*, which is a variety of *L. radiata* (L'Hér.) Herb. (red spider lily), from a commercial source (Heiwaen, Tenri, Nara Prefecture, Japan) and cultivated them in an experimental garden at Tokyo Metropolitan University (35.6° N, 139.4° E, 120 m above sea level; hereafter, TMU) from early June 2009 until mid-April 2010. With respect to cultivation, bulbs with a fresh mass of 2.0–3.0 g were planted individually in flexible plastic pots (13.5 cm in diameter, 11.5 cm deep) containing Kuroboku soil (an Andosol). The total N and the carbon : nitrogen (C : N) ratios of dried samples of the soil were approximately 5 g kg$^{-1}$ and 16, respectively. The pots were then buried in the experimental garden such that the pot's soil surface was at the same level as the ground surface. The plants received full sunlight almost all day and received natural precipitation. We did not irrigate or fertilize during the cultivation period. The air temperature at the highest point of the leaves (approx. 3 cm above the ground) and the soil temperature in the pots (at a depth of approx. 5 cm) were recorded at 1 h intervals with sensor-equipped data loggers (RT-30S, Espec MIC, Niwa, Aichi Prefecture, Japan), and the daily means were calculated. The daily mean soil temperature during the leafless period ranged from 19.8 to 30.8°C, and the daily mean air temperature during the growing period ranged from −2.3 to 21.0°C. The daily minimum air temperature was below zero for 76 days, reaching levels below −10°C for 7 days (electronic supplementary material, figure S1A). The annual precipitation from June 2009 to May 2010 at the nearest weather station, which is located approximately 8 km northwest of TMU, was 1418 mm (Japan Meteorological Agency). In mid-April 2010, the plants were removed from the pots and transferred to a laboratory at Nippon Medical School (hereafter NMS), which is approximately 25 km east of TMU and 8 m above sea level. Among the prepared plants, we chose two-leaf plants that had a fresh mass of 5.0–7.0 g for the fertilizer supply experiment and potted them individually into flexible plastic pots (10.5 cm in diameter and 12.0 cm deep) that contained fine (0.5– 2.0 mm) vermiculite and watered the substrate to approximately field capacity. The plants were placed in a laboratory until mid-June. All the plants had shed their leaves by mid-May and were leafless until early October. The plants remained juveniles without flowers throughout the study.

## 2.2. Fertilizer treatments during different seasons

In mid-June 2010, a total of 170 plants were assigned randomly to one of the following eight fertilizer treatments (the experimental protocol is shown in figure 1): fertilizer supplied in summer only (Su, $n = 21$), in autumn only (Au, $n = 21$), in winter only (Wi, $n = 22$), both in the summer and autumn (SA, $n = 21$) and both in the summer and winter (SW, $n = 22$). There were three control treatments that did not receive fertilizer (C0, C1 and C2; $n = 20, 21, 22$, respectively). C0 was used to determine the initial biomass and N content at the start of the fertilizer supply experiment; C1 was used as the control of Su, Au and SA; and C2 was used as the control of Wi and SW. The starting dates and the durations of the fertilizer supplies were 12 June 2010 and 45 days for the summer supply (Su, SA and SW), 8 October 2010 and 47 days for the autumn supply (Au and SA), and 7 January 2011 and 45 days for the winter supply (Wi and SW). During each fertilizer supply period, 45 ml of Hyponex 6-10-5 nutrient solution (Hyponex Japan, Osaka, Japan), which contained 0.71 mmol of N, 0.54 mmol of phosphorus and 0.21 mmol of potassium, as well as micronutrients, was applied to each plant weekly (six times in total). We weighed each pot after applying the nutrient solution and then watered the pots until they weighed 460 g (near field capacity of the vermiculite in the pot). On the same day, the other plants were also watered until the pots reached 460 g. When each fertilizer supply period ended, the vermiculite in the pots was replaced with fresh vermiculite that was not amended with nutrients to ensure that the plants could access the supplied nutrients only during the supply period. However, such treatment involves the potential risk of a disturbance that affects only plants supplied with fertilizer, especially those that received fertilizer for two periods (treatments SA and SW). To avoid such bias, we replaced the vermiculite of all the pots twice before the plants were harvested. The vermiculite was replaced in all the treatments after fertilizer application in the summer;

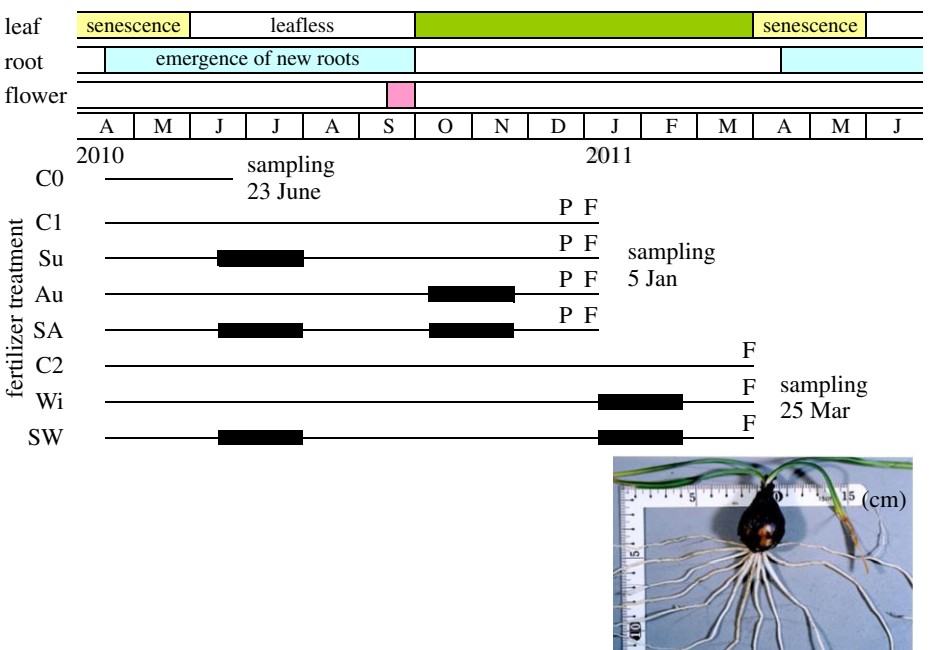

**Figure 1.** Phenology and morphology of *L. radiata* var. *radiata* and the protocol used for the fertilizer supply experiments. Potted plants were supplied with fertilizer during the periods shown by bold bars: summer only (Su), autumn only (Au), winter only (Wi), both summer and autumn (SA), and both summer and winter (SW). Treatments C0, C1 and C2 are controls without fertilizer application. Measurements of photosynthesis and chlorophyll *a* fluorescence were performed at the times indicated by P and F, respectively. A juvenile plant, which consists of a bulb (major storage organ), leaves and roots, is shown in the image. All plants used for the experiments were juveniles that lacked flowers.

in treatments Su, Au, SA and C1 after fertilizer application in the autumn; and in treatments Wi, SW and C2 after fertilizer application in the winter. Outside of the fertilizer supply periods, all the plants were watered weekly until the pots reached 460 g.

With the exception of those in C0, the pots were placed in 10 Styrofoam containers; each container contained 15 pots arranged in three rows and five columns. Fourteen pots (two for each treatment) were randomly assigned to a place in each container, excluding places in the middle (row 2, column 3). The remaining 10 pots (one pot each for Su, Au, SA and C1 and two pots each for Wi, SW and C2) were placed in the middle of a randomly selected container. The containers were then placed under indoor conditions until early September (during the leafless period) to avoid natural rainfall and to keep the soil temperatures in the pots from becoming extremely high during the summer. Beginning 6 September 2010, the containers were placed on the rooftop of a building at NMS, where the plants received full sunlight almost all day. When rain was expected, the plants were moved indoors in advance. We exchanged the position of each container daily whenever possible. The plants started unfolding their leaves in early October. The length of each leaf was measured weekly until mid-November and every two weeks thereafter until harvest. The air temperature at the highest points of the leaves and the soil temperature in the pots were continuously recorded. The daily mean soil temperature during the leafless period ranged from 17.1 to 32.3°C, and the daily mean air temperature during the growing period ranged from 3.1 to 24.9°C. The daily minimum air temperature was below zero for 13 days, and the minimum temperature was −2.8°C (electronic supplementary material, figure S1B).

## 2.3. Plant biomass and nitrogen content

The plants were harvested on 23 June 2010 (treatment C0), 5 January 2011 (treatments Su, Au, SA and C1), and 25 March 2011 (treatments Wi, SW and C2). Each individual plant was divided into the bulb, roots and leaves, which were then oven-dried for approximately 48 h at 80°C and then weighed. We determined the N concentration in each organ for six plants per treatment with an N–C analyser (Sumigraph NC80, Sumika Chemical Analysis Service, Osaka, Japan). To measure the leaf area, we made a photocopy of the leaves before drying them, and the scanned images were analysed with a raster graphics editor (Adobe Photoshop CS, Adobe Systems, CA, USA).

The dry mass of the individual plants at the start of the fertilizer supply experiment in mid-June (initial dry mass) was estimated from the fresh mass measured in mid-April using a linear relationship obtained from the C0 treatment, $D_{Ini} = 0.269F_{Apr} + 0.111$ ($n = 20$, $R^2 = 0.820$, $p < 0.001$), where $D_{Ini}$ and $F_{Apr}$ are the initial dry mass (g) and the fresh mass in mid-April (g), respectively. The bulb dry mass was assumed to be 95.7% of the total plant dry mass (mean value for C0), and the rest was considered root dry mass. For Wi, SW and C2, we also estimated the dry mass of individual plants in early January 2011 by their total leaf length using an exponential function obtained from the plants harvested in early January (treatments Su, Au, SA and C1), $D_{Jan} = 1.44e^{0.847L}$ ($n = 82$, $R^2 = 0.803$, $p < 0.001$), where $D_{Jan}$ and $L$ are the total plant dry mass (g) and total leaf length (m), respectively. The leaf area, $A$ (m$^2$), was also estimated from the total leaf length ($L$) as $A = 0.00493L - 0.000157$ ($n = 82$, $R^2 = 0.957$, $p < 0.001$), which was then converted to leaf dry mass by multiplying by the leaf mass per area (LMA). The mean LMA value for C1 (81.4 g m$^{-2}$) was used for Wi and C2 because these plants were treated equally until early January; likewise, the mean LMA value for Su (84.2 g m$^{-2}$) was used for SW. The bulb dry masses were assumed to be 84.5% (for Wi and C2) and 81.1% (for SW) of the total plant dry mass, which are the mean values for C1 and Su, respectively, and the remaining portion was considered root dry mass. The N content of each organ of each individual plant was obtained by multiplying the biomass and N concentration of the organ; for the N concentration of each organ, the mean values for each fertilizer treatment ($n = 6$) were used. For the estimation of the initial N content at the start of the fertilizer supply experiment, the mean N concentration values of C0 were used. The mean N concentration values of C1 and Su were used to estimate the early January N contents for C2 plus Wi and SW, respectively.

To examine the effect of seasonally absorbed N on leaf growth, we estimated the allocation ratio of the absorbed N in the leaves and the leaf growth supported by unit investment of N in the leaves. We estimated these values for the plants supplied with fertilizer during the summer and/or autumn (treatments Su, Au and SA); N absorbed during the winter was not invested in leaves (see the Results section). The amount of N that was absorbed during the fertilizer supply period and was invested in the leaves ($N_{leaf}$) was estimated as

$$N_{leaf} = (\text{leaf N content of the fertilized plant}) - (\text{mean leaf N content in C1 plants}).$$

Leaf growth (dry mass or area bases) supported by the N investment ($G_{leaf}$) was estimated as

$$G_{leaf} = (\text{leaf mass or area of the fertilized plant}) - (\text{mean leaf mass or area in C1 plants}).$$

These estimations used data from plants that were harvested in early January 2011 and whose N concentrations were determined for each organ (six plants per treatment); leaf N content was thus calculated using the individual leaf N concentrations of these plants. The allocation ratio of absorbed N in the leaves was obtained by dividing $N_{leaf}$ by the net N increase in the plant between mid-June and early January. The leaf growth supported by unit investment of the N in the leaves was calculated as $G_{leaf}/N_{leaf}$.

## 2.4. Photosynthesis measurements

Photosynthesis rates were measured in mid-December 2010 for Su, Au, SA and C1 using a portable photosynthesis measurement system (LI-6400, LI-COR, Lincoln, NE, USA) that contained a chamber (2 cm × 3 cm) equipped with a blue–red light source (6400-02B LED source). At the start of the measurement for each leaf, the chamber was set at 500 µmol m$^{-2}$ s$^{-1}$ of photosynthetic photon flux density (PPFD) and 36 Pa of ambient $CO_2$ partial pressure. Then, the PPFD was increased stepwise and relatively quickly to 1500 µmol m$^{-2}$ s$^{-1}$ (usually in 1–2 min for each of 500, 800 and 1200 µmol m$^{-2}$ s$^{-1}$) since our preliminary experiments showed that approximately 1200–1500 µmol m$^{-2}$ s$^{-1}$ of PPFD saturated the photosynthesis rate of *L. radiata* var. *radiata* plants cultivated in the TMU experimental garden. After the photosynthesis rate was confirmed to be stable at 1500 µmol m$^{-2}$ s$^{-1}$ of PPFD, the ambient $CO_2$ partial pressure was reduced to 0 Pa. The photosynthesis rate was determined under increasing ambient $CO_2$ partial pressures (0, 10, 20, 29, 36, 49, 59 and 119 Pa) at 1500 µmol m$^{-2}$ s$^{-1}$ of PPFD and at a leaf temperature of 22°C. The leaf temperature was chosen because we previously found for *L. radiata* var. *radiata* plants potted in the TMU experimental garden that the leaf temperature was 19.9–22.1°C under 1000–1500 µmol m$^{-2}$ s$^{-1}$ PPFD (the abovementioned values are the monthly means of the leaf temperature data recorded from December 2008 until March 2009 at 1 min intervals). For some plants in each fertilizer treatment, measurement was also performed at an ambient $CO_2$ partial pressure of 149 Pa, but the increase in the photosynthesis rate was only within 1% of the value at 119 Pa

in most of the plants. The initial slope of the photosynthetic intercellular $CO_2$ response curves was obtained by linear regression; data obtained at ambient $CO_2$ partial pressures of 0–29 Pa were used ($R^2 > 0.994$, $p < 0.005$). The initial slope indicates the leaf capacity of ribulose-1,5-biphosphate (RuBP) carboxylation, while the photosynthesis rate at the saturating ambient $CO_2$ partial pressure indicates the RuBP regeneration capacity [26]. We used the longest leaf of eight plants per treatment; the portion of each leaf that had developed by mid-November (during the period of leaf flush) was inserted into the chamber for measurements. We measured the width of the leaf portion and calculated the area. The obtained area-based net photosynthesis rates were converted to mass- and N-based net photosynthesis rates by dividing by LMA and area-based N concentrations, respectively. For the leaf N concentrations, the mean values for each fertilizer treatment ($n = 6$) were used.

## 2.5. Chlorophyll $a$ fluorescence measurements

In early January, the Chlorophyll $a$ (Chl$a$) fluorescence of intact leaves of 12 plants per treatment (treatments Su, Au, SA and C1) was measured with a portable Chl$a$ fluorometer (MINI-PAM; Heinz Walz, Effeltrich, Germany) connected to a leaf-clip holder (Model 2030-B). Measuring light and saturating light pulses (800 ms pulse of 7000–8000 µmol m$^{-2}$ s$^{-1}$ PPFD) were applied via a fibre-optic cable oriented 60° to the leaf plane and targeted to the centre of the portion of the leaf that had developed by mid-November. The distance between the leaf and fibre-optic cable was manually adjusted and maintained throughout the measurements. The effect of the internal temperature on the output of the measuring light LED was calibrated by assuming that a 1°C increase results in a 1% decrease in the measuring light intensity. Measurements of the PPFD on the leaf adaxial surface and the temperature on the leaf abaxial surface were made with a micro-quantum sensor and a Ni-CrNi thermocouple, respectively, which were attached to the leaf-clip holder.

We determined the minimal Chl$a$ fluorescence yield ($F_o$) and the maximal Chl$a$ fluorescence yield ($F_m$) of photosystem II (PSII) in the dark-adapted state when the plants were in a nearly unstressed state; the studied plants were kept indoors during the day before the measurements (air temperature = 10–15°C, PPFD < 10 µmol m$^{-2}$ s$^{-1}$), after which they were placed in an incubator (MIR-254, Panasonic Healthcare, Tokyo, Japan) set at 20°C in darkness for 12 h. These parameters were also measured on a morning following a sunny day. We also measured the steady-state Chl$a$ fluorescence yield ($F$) and the maximal Chl$a$ fluorescence yield ($F_m'$) in the light-adapted state of PSII on the second day of two consecutive sunny days. On day 1, after the $F_o$ and $F_m$ of the plants in a nearly unstressed state were determined in the morning, the plants were moved to the rooftop for exposure to the sun (the maximum PPFD and cumulative PPFD for the day were 1280 µmol m$^{-2}$ s$^{-1}$ and 21.3 mol m$^{-2}$, respectively). To investigate the effects of freezing night temperature on the photochemical capacity during the following day, half of the plants ($n = 6$ for each treatment) were exposed to a minimum −8°C air temperature during the night in an incubator (MIR-254); the temperature in the incubator was 5°C initially, after which it was lowered gradually to −8°C for 4 h, maintained at −8°C for 4 h, and then increased gradually to 5°C for 4 h. Because the pots were covered with insulating material, the vermiculite in the pots did not freeze. The other half of the plants were kept in a cold room (5°C) overnight. On the morning of day 2, we measured the $F_o$ and $F_m$ of the plants at 20°C, after which we moved the plants to the rooftop for exposure to the sun. Measurements of $F$ and $F_m'$ were performed three times per plant within 1.5 h beginning at 09.30; for the calculation of the photochemical parameters mentioned below, we used only data obtained when the PPFD > 1100 µmol m$^{-2}$ s$^{-1}$ (the maximum PPFD was 1348 µmol m$^{-2}$ s$^{-1}$). The leaf portions for pigment analyses were collected immediately after these measurements were performed. In late March 2011, we also measured the $F_o$ and $F_m$ of plants that were in a nearly unstressed state in treatments C2, Wi and SW.

The maximum quantum yield of PSII ($F_v/F_m$) was calculated as $(F_m − F_o)/F_m$. The following parameters for the light-adapted state of PSII were also calculated: the effective quantum yield of PSII ($\Delta F/F_m' = (F_m' − F)/F_m'$); photochemical quenching ($q_P = (F_m' − F)/(F_m' − F_o')$); the quantum yield of open PSII reaction centres ($F_v'/F_m' = (F_m' − F_o')/F_m'$); and non-photochemical quenching (NPQ $= (F_m − F_m')/F_m'$), where $F_o'$ is the minimum fluorescence yield in the light-adapted state and is calculated as $F_o/(F_v/F_m + F_o/F_m')$ according to a previous report [27]. For the calculation of $F_o'$, the $F_o$ and $F_m$ of plants in a nearly unstressed state were used.

## 2.6. Pigment analyses

From each plant used for the measurement of Chl$a$ fluorescence, we collected 3 cm long leaf portions, which included the measured point in the centre and measured their widths. The leaf samples were immediately

frozen in liquid N and stored at −80°C until the analyses. To prepare leaf extracts, each leaf sample was ground with a mortar and pestle, after which the pigments were extracted with 2 ml of acetone : methanol (7 : 2, by volume) that contained 10 mM Tris-HCl (pH 8.0) to prevent acidification. Preliminary tests confirmed that almost all the pigments were extracted during the first extraction; thus, we skipped additional extractions. As an internal standard, 50 µl of canthaxanthin (Roche, Basel, Switzerland) dissolved in chloroform : ethanol (2 : 1, by volume) was added to the extracts. The concentration of canthaxanthin was determined by its absorbance in benzene [28]. The extract was then filtered through a syringe filter (Chromatodisc 4 N, 0.45 µm pore size, Kurabo, Osaka, Japan) and kept in the dark at approximately 2°C until analysis (usually within 30 min). The pigments in the leaf extracts were analysed with an high performance liquid chromatography (HPLC) apparatus equipped with a Novapak $C_{18}$ column (8 × 100 mm, RCM type, Waters, Milford, MA, USA) and a photodiode array detector (MCPD-3600, Otsuka Electronics, Osaka, Japan). The HPLC procedures followed those of [29]. Approximately 50 µl of the leaf extracts that contained canthaxanthin as an internal standard was injected into the HPLC system and then eluted with 1.75% methanol, 1.75% dichloromethane, 1.75% water and 94.75% acetonitrile (by volume) for 8 min at a flow rate of $2.0\ \mathrm{ml\ min^{-1}}$ at room temperature, followed by 50% acetonitrile and 50% ethyl acetate (by volume) for 12 min. Each pigment was identified by its retention time and absorption spectrum. The pigments were quantified in terms of the peak area (absorbance units × min) at 440 nm. Standard pigments were isolated and purified from spinach (*Spinacia oleracea* L.) leaves. The molar extinction coefficients of carotenoids (in ethanol) and chlorophyll (in methanol) were obtained from [28] and [30], respectively. The obtained area-based pigment concentrations were converted to mass- and N-based pigment concentrations by dividing by LMA and area-based N concentrations, respectively. For the leaf N concentrations, the mean values for each fertilizer treatment ($n = 6$) were used.

## 2.7. Statistical analyses

We used R v. 3.3.1 software for Windows [31] for statistical analyses. To determine the effects of the fertilizer treatments, we used Games–Howell's method for multiple comparisons (http://aoki2.si. gunma-u.ac.jp/R/src/tukey.R). A paired-sample *t*-test was used to detect temporal changes in the N content (http://aoki2.si.gunma-u.ac.jp/R/src/paired_t_test.R), and Welch's *t*-test was used to test for differences in photochemical parameters between the plants exposed to two different night-time temperatures (http://aoki2.si.gunma-u.ac.jp/R/src/indep_sample.R). Correlations between leaf area and plant growth (change in total plant biomass) were detected with the Pearson product-moment correlation coefficient (http://aoki2.si.gunma-u.ac.jp/R/src/my_cor.R).

# 3. Results

## 3.1. Nitrogen uptake and allocation

All fertilizer treatments significantly increased the plant N contents (figure 2*a*), indicating that the plants absorbed N during the treatment periods. In C2, despite no fertilizer treatment, an increase in plant N content was detected. This finding suggests that there was some extra N supplied, which we have not determined at present. However, the plants supplied with fertilizer in the winter (treatments Wi and SW) exhibited a greater increase in plant N than did the plants in C2, indicating that the plants absorbed the applied fertilizer during the winter. A clear seasonal trend was detected in the allocation of N (figure 2*b*), as N was allocated mainly to the leaves and roots until early January. By contrast, N was allocated exclusively to the bulb, the major storage organ, during the later part of the growing period (January–March).

## 3.2. Effects of the timing of nitrogen uptake on leaf growth and net biomass production

Fertilizer application during the summer enhanced leaf growth; both the number of leaves and the total leaf length were significantly greater for plants supplied with fertilizer in the summer (Su, SA and SW) than for the other plants (figure 3). In response to fertilizer application in autumn, leaf number increased only marginally; although the leaf length increased significantly, the extent of the increase was smaller than the response to the summer application (compare treatments Au and Su in figure 3*b*). This result could be explained in part by the lower uptake of N (figure 2*a*) and the lower allocation ratio of absorbed N in the leaves (figure 4*a*) of the Au plants than that of the Su plants. In addition, the Au plants exhibited weaker effects of absorbed N on leaf mass and area growth than the Su plants

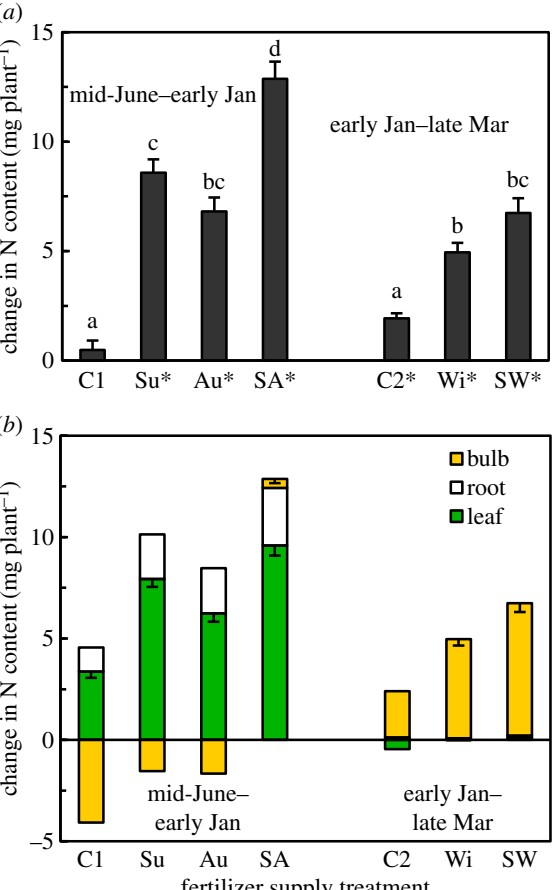

**Figure 2.** Net nitrogen (N) uptake and N allocation to each organ of *L. radiata* var. *radiata* plants treated with fertilizer at different times (figure 1). Changes in the N content in (*a*) a whole plant and in (*b*) each organ of a plant during the early (from mid-June 2010 to early January 2011) and the late (from early January to late March 2011) parts of the experimental period. The N content of each organ of each individual plant was obtained by multiplying the biomass and N concentration of the organ; for the N concentration of each organ, the mean value for each fertilizer treatment ($n = 6$) was used. For the estimation of the N content in mid-June, the mean N concentrations for C0 were used. The mean N concentrations of C1 and Su plants were used to estimate the early January N contents of C2 plants plus Wi and SW, respectively. The error bars denote 1 s.e.m. ($n = 20$–22); they are not shown when 1 s.e.m. $< 0.2$. In (*a*), different lowercase letters above the bars represent significant differences among the fertilizer supply treatments ($p < 0.05$, Games–Howell's method for multiple comparisons). Asterisks (*) besides the treatment code indicate that a significant change in the plant N content occurred (i.e. the plants took up N) during the experimental period ($p < 0.05$, paired-sample *t*-test).

(figure 4*b*), meaning that N uptake during the autumn contributed more to the increase in leaf N concentration than that during the summer.

Net biomass production (i.e. change in biomass) was positively correlated with leaf area (figure 5*a*) and thus was significantly greater in the plants supplied with fertilizer in the summer (treatments Su, SA and SW) than in the other plants (figure 5*b*). Those plants supplied with fertilizer in the summer also exhibited a greater biomass increase per unit leaf area in the earlier part of the growing period (until early January), whereas such superiority was not detected in the later part of the growing period (from early January to late March, figure 5*c*).

## 3.3. Effects of the timing of nitrogen uptake on photosynthetic capacity

Fertilizer application in the summer and autumn (treatments Su, Au and SA) significantly increased the LMA, leaf area- and mass-based concentrations of N, chlorophyll and carotenoids and photosynthetic capacities of the plants (figure 6). The effects were greater when the fertilizer was supplied in the autumn (treatment Au) rather than in the summer (treatment Su). The initial slope of the photosynthetic intercellular $CO_2$ response curves was greater in all three fertilizer treatments than in

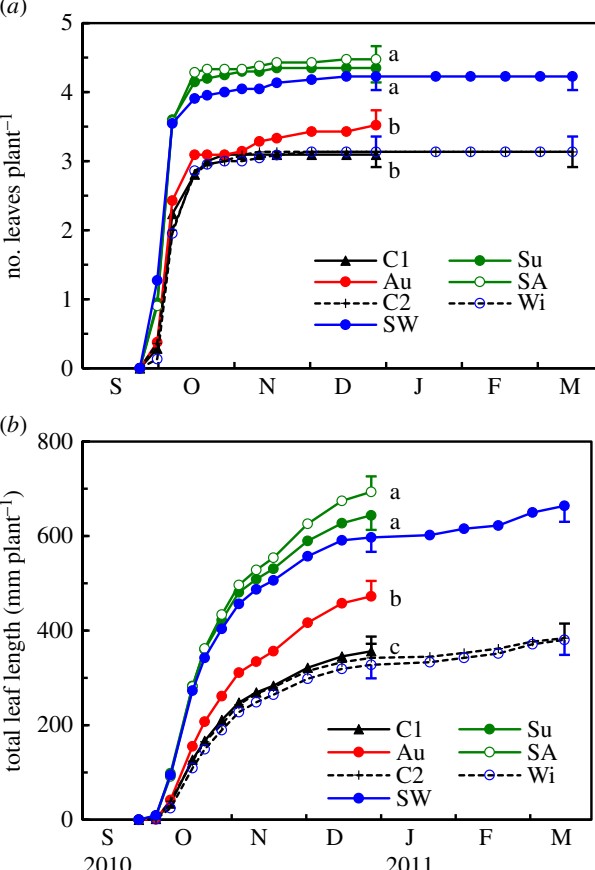

**Figure 3.** Leaf growth of *L. radiata* var. *radiata* plants treated with fertilizer at different times (figure 1). Seasonal changes in (*a*) the number of leaves and (*b*) the total leaf length. The error bars denote 1 s.e.m. ($n = 20$–22). Different lowercase letters represent significant differences among the fertilizer supply treatments in late December 2010 ($p < 0.05$, Games–Howell's method for multiple comparisons). For the statistical analyses, the data for Su and SW and those for Wi, C1 and C2 were pooled together, as they were treated equally until early January 2011.

the control C1 (figure 6*j,k*). However, the photosynthesis rates at near saturating ambient $CO_2$ partial pressure (119 Pa) increased only in response to the fertilizer application in the autumn (treatments Au and SA, figure 6*m,n*).

   The effective quantum yield of PSII ($\Delta F/F'_m$), photochemical quenching ($q_P$) and the quantum yield of open PSII reaction centres ($F'_v/F'_m$) under high irradiance conditions (1100 < PPFD < 1348 µmol m$^{-2}$ s$^{-1}$) were all highest in the plants supplied with fertilizer only in autumn (Au), suggesting that these plants had the greatest photochemical ability among all the treatments (figure 7*a–c*). Application of fertilizer in summer only (treatment Su) had no significant effect on photochemical ability. After being exposed to freezing temperatures (minimum of −8°C) at night, the plants in all the treatments exhibited increased NPQ (an indicator of thermal dissipation of excess light energy, which protects PSII) values (figure 7*d*). The NPQ was lowest in the Au plants, although the differences among the treatments were not statistically significant. Across all the treatments, the $F_v/F_m$ was maintained at values greater than 0.7 even after plants experienced a night with freezing temperature (table 1), indicating no chronic impairment of PSII. The leaf N-based photosynthetic capacities were highest in C1 (figure 6*l,o*).

## 4. Discussion

The present study clearly showed a shift in the function of seasonally absorbed N in *L. radiata* var. *radiata* from the increase in leaf growth in response to N absorption during the leafless summer period to the increase in leaf N concentration and hence photosynthetic capacity in response to N absorption in early autumn (the period of leaf flush). This functional shift is considered to be rational in terms of effectively increasing the growth and survival of this winter-green herb. From autumn to early winter,

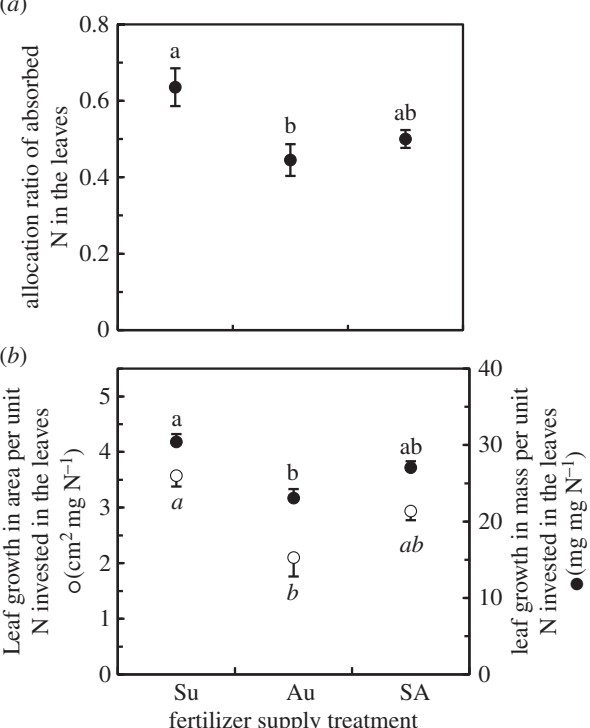

**Figure 4.** Estimated effects on *L. radiata* var. *radiata* leaf growth in response to nitrogen (N) absorbed during the different fertilizer supply periods (figure 1): (*a*) allocation ratio of absorbed N in the leaves, (*b*) leaf growth in area (open circles) and in mass (closed circles) supported by unit investment of the N in the leaves (for detailed explanation of the method of estimation, see the Materials and methods section). The error bars denote 1 s.e.m. (*n* = 6). Different lowercase letters represent significant differences among the fertilizer supply treatments ($p < 0.05$, Games–Howell's method for multiple comparisons).

the plants supplied with fertilizer in the summer (Su) produced more biomass than did those supplied with fertilizer in the autumn (Au) (figure 5*b*). The greater increase in biomass of Su plants was achieved not only by their greater leaf area (figure 5*a*) but also by their greater biomass increase per unit leaf area (figure 5*c*) despite their lower area-based photosynthetic capacity (figure 6*j,m*). This result provides clear evidence that investing the N absorbed before the growing season (i.e. stored N) in an increase in leaf quantity effectively promoted growth. Such an investment enabled the rapid development of a large amount of leaves (figure 3). In addition, the leafy plants probably paid reduced C costs per unit leaf area associated with respiration of non-photosynthetic organs, which would also have positively contributed to a net biomass increase [32,33].

By contrast, N absorption in the autumn enhanced leaf N concentrations and capacities for RuBP-carboxylation (figure 6*j,k*) and regeneration (figure 6*m,n*), which means that a large amount of the N absorbed in autumn was invested in relevant photosynthetic proteins. Such an investment of N would be essential for the survival and growth of *L. radiata* var. *radiata* during winter. In winter, the senescence of leaves of deciduous species creates high light availability for photosynthesis of winter-green plants. However, excess light can cause a potentially fatal risk. High light availability combined with low winter temperature (which reduces the activity of photosynthetic enzymes) leads to an imbalance between the absorption and use of photons. The resulting excessive amount of photons generates reactive oxygen species (photo-oxidative stress), which impair leaf tissues and cause plant death in the most severe cases [34,35]. *Lycoris radiata* var. *radiata* is thought to combat this fatal risk by maintaining photosynthetic activity (thus maintaining the use of photons) even under low winter temperatures. This is an overwintering strategy reported in other species that grow intensively during winter, such as spinach and winter wheat [36–39]. We expect that the high photosynthetic capacity of Au plants would lead to greater growth per unit leaf area than that observed in other treatments in mid-winter (data for C2, Wi and SW plants are shown in figure 5*c*) because photosynthetic capacity is thought to determine plant growth more directly in mid-winter (compared with autumn to early winter), when plants have already established foliage, and the C costs associated with respiration of non-photosynthetic organs would be low owing to the low growth temperature. The Au plants had high LMA values (figure 6*a*), which also suggests their superior performance under low temperature

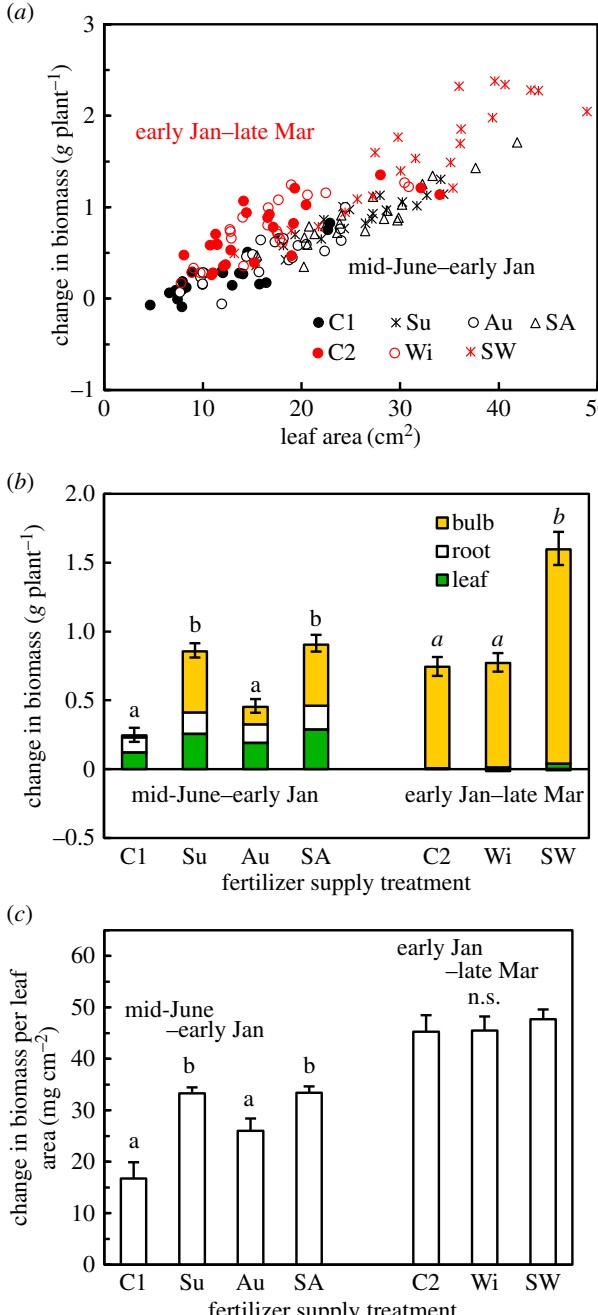

**Figure 5.** Growth of *L. radiata* var. *radiata* plants treated with fertilizer at different times (figure 1) during the early (from mid-June 2010 to early January 2011) and late (from early January to late March 2011) parts of the experimental period: (*a*) correlations between leaf area and change in total plant biomass ($r = 0.950$, $n = 82$, $p < 0.001$, and $r = 0.913$, $n = 65$, $p < 0.001$ for the early and the late parts of the experimental period, respectively), (*b*) change in the biomass of each organ and (*c*) change in total plant biomass per leaf area. Leaf area values on 18 November 2010 and on 25 March 2011 were used for the early and late parts of the experimental period, respectively. Different lowercase letters above the bars in (*b*) and (*c*) represent significant differences among the fertilizer supply treatments within each experimental period ($p < 0.05$, Games–Howell's method for multiple comparisons). The error bars denote 1 s.e.m. ($n = 20$–$22$). In (*b*), the standard errors outside and inside the bars correspond to the whole plant and bulb, respectively. The values of 1 s.e.m. were smaller than 0.02 for the root and leaf and are not shown.

and high light conditions [40]. The plants probably had thick laminae, which contribute to high photosynthetic capacity under high irradiance. In addition, the plants may have had rigid cell walls, which contribute to freezing tolerance [41].

A lack of N investment in leaves during winter is also rational because growth temperature increases towards spring, after which the growing season ends. Instead, N was stored in the bulb (figure 2*b*), which

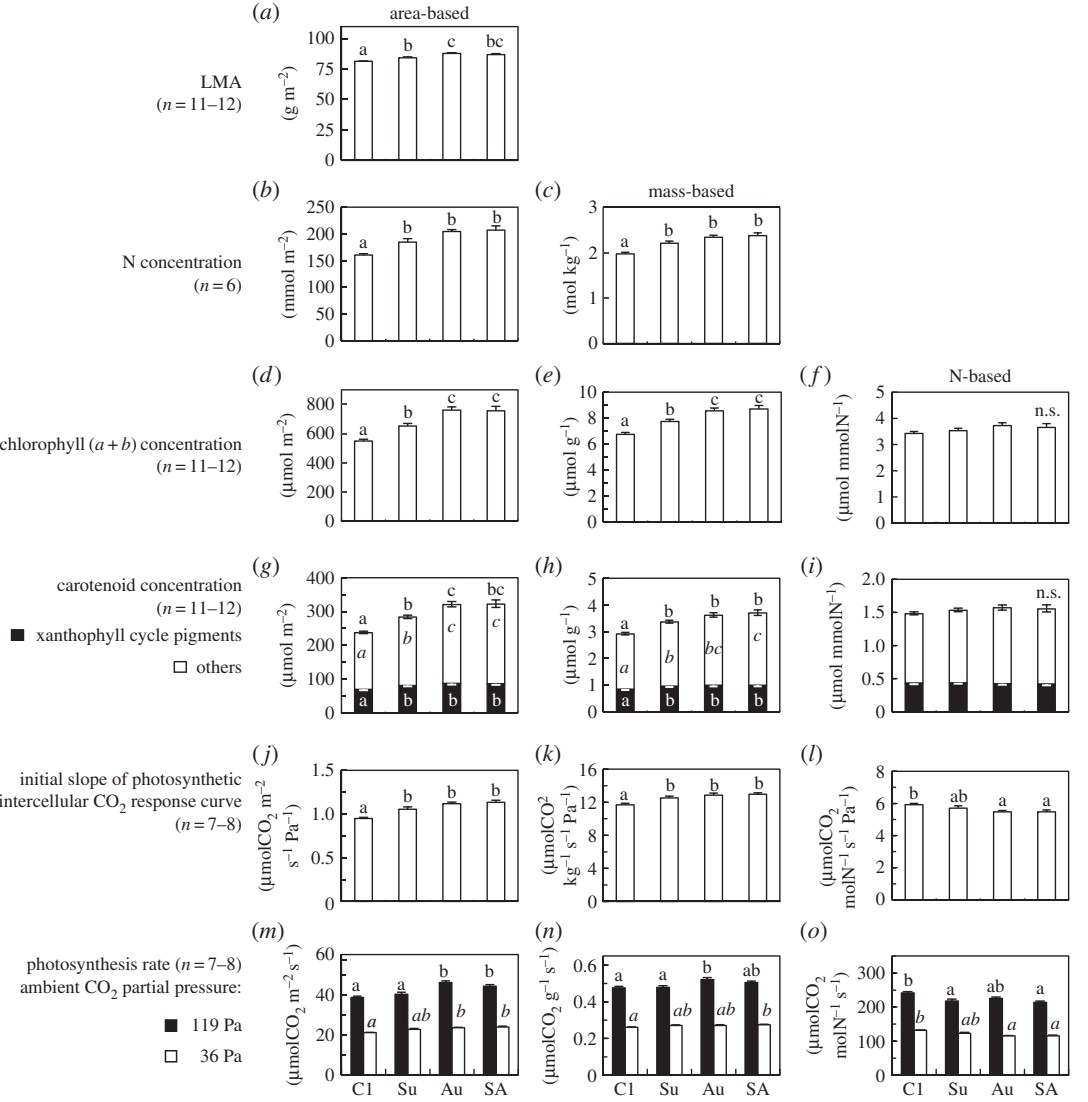

**Figure 6.** Leaf characteristics of *L. radiata* var. *radiata* plants treated with fertilizer at different times (figure 1): (*a*) LMA; concentrations of (*b*,*c*) nitrogen (N), (*d*–*f*) chlorophyll and (*g*–*i*) carotenoids; and (*j*–*o*) photosynthetic capacities. The photosynthesis measurements were conducted in mid-December 2010 at 22°C and a PPFD of 1500 µmol $m^{-2}$ $s^{-1}$. Other data were obtained from samples harvested in early January 2011. The panels represent leaf area-based (left), leaf mass-based (middle) and leaf N-based (right) parameters. The error bars denote 1 s.e.m. For the carotenoid concentrations, the standard errors outside the bars correspond to the total measured carotenoids, and those inside the bars correspond to the total xanthophyll cycle pigments (violaxanthin, antheraxanthin and zeaxanthin) and that of the others (neoxanthin, lutein and β-carotene). Different lowercase letters represent significant differences among the fertilizer supply treatments ($p < 0.05$, Games–Howell's method for multiple comparisons).

will contribute to growth during the next season, probably by enhancing leaf growth, such as N absorption in summer, or via effects on root production during the leafless period. The N absorbed during winter is not the only N that is stored in the bulb. Portions of N that are absorbed in summer and autumn and allocated to leaves are also translocated during leaf senescence and stored in the bulb. The percentage of leaf N that is translocated to storage may depend on the time of absorption, as shown in *Quercus serrata* saplings [14]. The contribution of the stored N to the growth during the next season may also vary depending on the time of original absorption and the time of storage. These are important topics that need to be examined to evaluate the effect of nutrient seasonality on plant performance.

An annual N-use schedule similar to that of *L. radiata* var. *radiata* has been reported for the understory shrub *Aucuba japonica* in Japan [7,22,42]. This evergreen shrub species develops new leaves during early

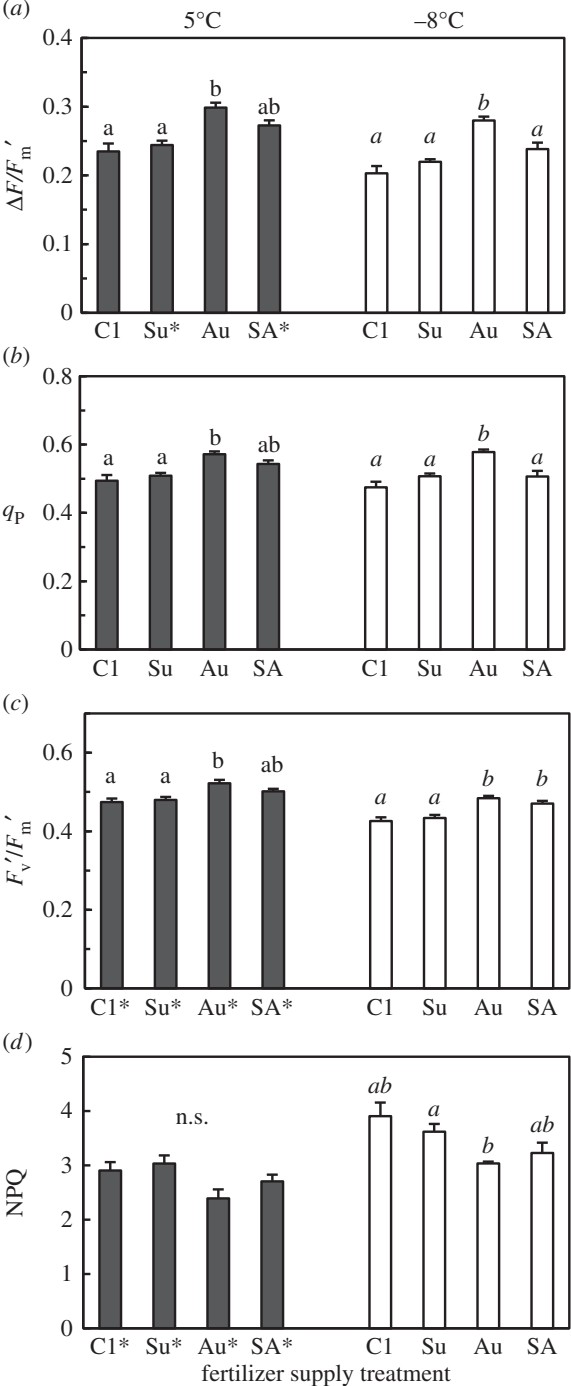

**Figure 7.** Photochemical properties of PSII in *L. radiata* var. *radiata* plants treated with fertilizer at different times (figure 1): (*a*) effective quantum yield of PSII $(\Delta F/F'_m)$, (*b*) photochemical quenching ($q_P$), (*c*) quantum yield of open PSII reaction centres $(F'_v/F'_m)$ and (*d*) non-photochemical quenching (NPQ). The parameters were determined before noon under 1100–1348 µmol m$^{-2}$ s$^{-1}$ of PPFD; plants that had been exposed to freezing ($-8°C$ was the minimum) or relatively mild (5°C) temperatures during the night before the measurement day were used. The error bars denote 1 s.e.m. ($n = 6$). Different lowercase letters above the bars represent significant differences among the fertilizer supply treatments under the same night temperature conditions ($p < 0.05$, Games–Howell's method for multiple comparisons). Asterisks (*) beside the treatment code indicate a significant difference in the property between the two night-time temperature conditions for each fertilizer treatment (Welch's $t$-test, $p < 0.05$).

summer, which are supported by the massive reallocation of N stored within old leaves. The leaf area per plant is nearly constant during the rest of the year; however, the leaf N concentration increases from autumn to winter and then decreases towards early summer. It has been shown that the seasonal trend in leaf N concentrations in this shrub species is parallel to that of the estimated optimum, which

**Table 1.** Maximum quantum yield ($F_v/F_m$) of PSII in the dark-adapted state in *L. radiata* var. *radiata* plants treated with fertilizer at different times (figure 1). (Measurements were performed on plants in a nearly unstressed state (see the Material and methods section) in early January and late March. In January, measurements were also performed on plants that had been exposed to a sunny day followed by a night with freezing ($-8°C$ was the minimum) or relatively mild ($5°C$) temperatures. The values are means, with 1 s.e.m. in parentheses. No significant difference in $F_v/F_m$ was observed among the fertilizer treatments ($p > 0.05$, Games–Howell's method for multiple comparisons).)

| fertilizer supply treatment | early January | late March | | | | | |
| --- | --- | --- | --- | --- | --- | --- | --- |
| | C1 | Su | Au | SA | C2 | Wi | SW |
| nearly unstressed state ($n = 12$) | 0.780 (0.004) | 0.783 (0.002) | 0.789 (0.003) | 0.788 (0.005) | 0.782 (0.005) | 0.786 (0.005) | 0.798 (0.003) |
| after a sunny day followed by | | | | | | | |
| a mild night ($n = 6$) | 0.719 (0.013) | 0.730 (0.004) | 0.733 (0.005) | 0.732 (0.007) | | not measured | |
| a freezing night ($n = 6$) | 0.701 (0.008) | 0.706 (0.005) | 0.711 (0.007) | 0.700 (0.008) | | not measured | |

maximizes photosynthetic production per unit leaf N (N-use efficiency, see [23,24]) in response to the seasonally changing temperature and light conditions of the shrub's habitat [22].

Another notable finding in relation to the overwintering strategy of *L. radiata* var. *radiata* is that the leaf quality of the unfertilized control plants was high enough for the plants to survive and grow under at least the relatively mild winter conditions to which they were exposed during the experiment (electronic supplementary material, figure S1B). Neither leaf necrosis nor plant death was observed until late March. In addition, the increase in leaf area-based biomass of those control plants from early January to late March was as high as that of the plants supplied with fertilizer both in the summer and winter (SW plants, figure 5*c*), despite the lower leaf N concentrations in the former (figure 6*b*; see the electronic supplementary material, table S1 for leaf N concentrations in late March); i.e. the control plants exhibited higher N-use efficiency. The fact that the fertilizer application during the leafless summer period increased the leaf growth to a greater extent than that during the autumn (figure 3) suggests that the amount of N that is stored in the bulb before the growing season is a major determinant of leaf growth in *L. radiata* var. *radiata*. In addition, the stored N also determined the leaf N concentration in the case of the control plants, as these plants had no external N supply. The fairly good performance of the control plants therefore indicates that the quantity of leaves determined by the stored N is regulated to ensure a certain N concentration for the leaves during winter, even in years with no N availability during the growing season. This regulation is probably based on the evolutionary experience of the species in terms of year-to-year fluctuations both in seasonal N availability and the harshness of winter conditions, assuring long-term survival and growth through the winter. A contrasting result has been reported in two invasive species: *Acer pseudoplatanus* [15] and *Calamagrostis epigejos* [18]. These species, like *L. radiata* var. *radiata*, enhanced leaf production in response to increased levels of stored N; however, many of the leaves died later in the growing season owing to low N availability under the experimental conditions.

The significance of storage organs for the growth of perennial plants is well documented. Many studies have quantified how much of the N required for new growth is supplied by stored and external N [4,6–11]. In those studies, a common assumption was that the N absorbed during different seasons was complementary and satisfied the high N demand of the new growth. However, our results suggest that stored N does not fulfil the same functions as the N that is externally absorbed during the growing season. This phenomenon has important implications for estimating plant responses to environmental changes. Anticipated global changes involve altered seasonality (both temperature and precipitation; [43]), which, combined with increasing anthropogenic N deposits [44,45], would affect seasonal microbial activity and the resulting N availability for plants [46,47]. On the basis of the insight obtained in this study, we expect that these changes will strongly impact the performance of perennial plants that have large storage organs via altered relative investment of N into different functions.

Data accessibility. The datasets supporting this article have been uploaded as part of the electronic supplementary material.

Authors' contributions. All authors contributed to designing the experiments; S.N. and T.N. conducted the experiments; S.N. and A.I. analysed the data; S.N., A.I. and N.K. contributed to the writing of the manuscript.

Competing interests. We declare we have no competing interests.

Funding. This work was supported by Japan Society for the Promotion of Science KAKENHI Grant (grant no. 21570025 to S.N. and grant nos 16H02708 and 18H04149 to A.I.).

Acknowledgements. We thank Takehiro Masuzawa, Emiko Maruta and Yota Yokoi for their helpful advice. We also thank Eitetsu Sugiyama and Takako Yasuki for their assistance in the experimental garden at Tokyo Metropolitan University.

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
