## [Reviewer comments · Royal Society Open Science]

Review History

RSOS-190034.R0 (Original submission)

Review form: Reviewer 1

Is the manuscript scientifically sound in its present form?

Yes

Are the interpretations and conclusions justified by the results?

Yes

Is the language acceptable?

Yes

Is it clear how to access all supporting data?

Yes

Do you have any ethical concerns with this paper?

No

Have you any concerns about statistical analyses in this paper?

I do not feel qualified to assess the statistics

Recommendation?

Accept as is

Comments to the Author(s)

This is a well-written paper containing interesting result which merit publication. Carefully designed experiments produced the findings that elucidated the significance of the wintergreen habit of plants. I have no serious criticism regarding methodology, results and interpretation of results.

Review form: Reviewer 2**Is the manuscript scientifically sound in its present form?**

Yes

Are the interpretations and conclusions justified by the results?

Yes

Is the language acceptable?

Yes

Do you have any ethical concerns with this paper?

No

Have you any concerns about statistical analyses in this paper?

Yes

Recommendation?

Major revision is needed (please make suggestions in comments)

Comments to the Author(s)

General comments:

The manuscript is describing that nutrient absorbed during different season was allocated to different organs, and thus affects plant growth in a winter-green perennial herb. The research question is thus in principle of relevance to our understanding the nutrient use strategy of perennial plants. However, there are several issues with the manuscript, which should be addressed:

The introduction could use some more detail and more references to previous works. For example, instead of "... studies have shown different fates of N absorbed at different times ..." (P.3, L.55-56), the reader would be more interested in what kind of differences are there, as well as how and why these differences affect plant's "performance". It would be beneficial if a (or several) specific hypothesis were formulated. This would also make the discussion clearer.

The description of the methods lacks detail in several aspects (please refer to the detailed comments below). I would like to see a more thorough discussion. I think there are several other points worth discussing. For example, besides higher leaf nutrient concentration, the Au treatment leads to significantly higher LMA, which is considered as adaptation to high light availability and low temperature. Is it means that the plants receive Au treatment would perform better than plants in other treatments during the coldest period (Jan to Feb)?

See my detailed comments below:

Detail comments:

Introduction

P.3 L.45. here you mentioned “efficient uptake” (also in the very first sentence of abstract). I don’t think you really analyzed nutrient uptake process or efficiency. Maybe you should focus on seasonality.

P.3 L.51. See also Tully 2013 for the driving effect of nutrient demand in plants using internal nutrients. (Tully KL, Wood TE, Schwantes AM, et al. (2013). Soil nutrient availability and reproductive effort drive patterns in nutrient resorption in *pentaclethra macroloba*. Ecology 94:930-940.)

P.3 L.61. Here you mention that no study have examined the effect of seasonally absorbed N in wild plant species. However, your plant materials is obtained from commercial source (P.5 L.102.). Is it a cultivated variety?

P.3 L.63. define “plant performance”

P.3 L.65. There should be a “.” after the “var”.

P.3 L.65. “Amaryllidacease” should be “Amaryllidaceae”.

P.4, L.85-96. The reader expects concisely formulated research questions – or hypotheses here. For example, what kind of function would you expect with the summer/autumn/winter absorbed N, and Why.

Methods

P.7, L.163. no water supply out of the fertilizer supply period?

P.8, L.173. Why do you choose these dates to harvest the plants? Why harvest the C1, Su, Au, SA treatment at 5th January? It seems that the leaf is still growing (Figure 3B.)

Besides, why not harvest the plants in all the treatment at the same time after the leaf senescence? Figure 1 indicates that juvenile plants have leaves from September to next April. After that, the plants shed the leaves, left the storage organ (i.e. bulbs and maybe roots) only. During the leaf senescence, portions of the nutrients will be re-translocated from leaves to storage organs. The energy and nutrient stored in these organs affects the plants future performance. Thus, compare the biomass and the nutrient pool size of the storage organ among treatments at the end of the growing season might be a useful to evaluate the effect of nutrient seasonality on plant performance. I noticed that you would like to examine the effects on leaf “quality”, and some of the measurements might be destructive. However, there are over 20 replicates in each treatment. I think it is possible to test the differences with several replicates.

P.9, L.203. What does the Gleaf stands for? “Leaf amount” in the equation indicates biomass?

P.9, L.203. How do you decide the near-saturating PPF_D without measuring the light response curve?

P.13, L.306-313. The Games-howell’s method is a non-parametric post-hoc test. Did you perform any homogeneity test? If the data is of equal variance, will parametric test, such as TukeyHSD, achieve a higher statistical power?

P.13, L.306-313. What correlation analysis method was used in figure 5a?

Results:

P.13, L.330. Please consider move the sentences like "The marginal increase in the number of leaves in plants supplied with fertilizer only in autumn (treatment Au) suggests that" to the discussion section.

Discussion

P.16, L. 376. Please define "leaf quality". The term "leaf quality" is obscure, since many parameters, such as N content, LMA, photosynthetic parameters and etc., is significantly different between treatment with nutrient in autumn and other seasons.

P.16, L. 389. Please explain which parameters related with "RuBP- carboxylation and regeneration"

P.17, L395. Remove the trailing "." in "Lycoris."

P.18, L431. "radata" should be "radiata".

Figures

p. 26, Line 611 caption of figure 2. n = 20-22. However, in the methods section, line 177, you mentioned "determined the N concentration in each organ for six plants per treatment." Thus, the replication should be 6? There seem to be some contradictions.

p. 31-35, figures 2-6. Some of the characters in the units of the axis label should be superscripted or subscripted, such as "2" in "cm²" or "CO₂".

Decision letter (RSOS-190034.R0)

03-Jul-2019

Dear Dr Nishitani,

The editors assigned to your paper ("Functional differences in seasonally absorbed nitrogen in a winter-green perennial herb") have now received comments from reviewers. We would like you to revise your paper in accordance with the referee and Associate Editor suggestions which can be found below (not including confidential reports to the Editor). Please note this decision does not guarantee eventual acceptance.

Please submit a copy of your revised paper before 26-Jul-2019. Please note that the revision deadline will expire at 00.00am on this date. If we do not hear from you within this time then it will be assumed that the paper has been withdrawn. In exceptional circumstances, extensions may be possible if agreed with the Editorial Office in advance. We do not allow multiple rounds of revision so we urge you to make every effort to fully address all of the comments at this stage. If deemed necessary by the Editors, your manuscript will be sent back to one or more of the original reviewers for assessment. If the original reviewers are not available, we may invite new reviewers.

To revise your manuscript, log into <http://mc.manuscriptcentral.com/rsos> and enter your

Author Centre, where you will find your manuscript title listed under "Manuscripts with Decisions." Under "Actions," click on "Create a Revision." Your manuscript number has been appended to denote a revision. Revise your manuscript and upload a new version through your Author Centre.

- Data accessibility

If you wish to submit your supporting data or code to Dryad (<http://datadryad.org/>), or modify your current submission to dryad, please use the following link:
<http://datadryad.org/submit?journalID=RSOS&manu=RSOS-190034>

- Competing interests

- Authors' contributions

- Acknowledgements

- Funding statement

Kind regards,

Alice Power

Editorial Coordinator

on behalf of Kevin Padian (Subject Editor)

Associate Editor's comments:

The Editors have made their decision on the basis of their read of the paper and the comments of reviewer 2. As you will see, the second referee is more critical of your paper, but nevertheless sees some merit in further consideration. Please ensure you fully respond to this reviewer's concerns, and incorporate the required changes into your manuscript before you submit. We would also ask that you supply a full point-by-point response to each query/comment of the reviewer and how you've tackled these. Good luck.

Comments to Author:

Reviewers' Comments to Author:

Reviewer: 1

Comments to the Author(s)

This is a well-written paper containing interesting result which merit publication. Carefully designed experiments produced the findings that elucidated the significance of the wintergreen habit of plants. I have no serious criticism regarding methodology, results and interpretation of results.

Reviewer: 2

Comments to the Author(s)

General comments:

The manuscript is describing that nutrient absorbed during different season was allocated to different organs, and thus affects plant growth in a winter-green perennial herb. The research question is thus in principle of relevance to our understanding the nutrient use strategy of perennial plants. However, there are several issues with the manuscript, which should be addressed:

The introduction could use some more detail and more references to previous works. For example, instead of "... studies have shown different fates of N absorbed at different times ..."

(P.3, L.55-56), the reader would be more interested in what kind of differences are there, as well as how and why these differences affect plant's "performance". It would be beneficial if a (or several) specific hypothesis were formulated. This would also make the discussion clearer.

The description of the methods lacks detail in several aspects (please refer to the detailed comments below). I would like to see a more thorough discussion. I think there are several other points worth discussing. For example, besides higher leaf nutrient concentration, the Au treatment leads to significantly higher LMA, which is considered as adaptation to high light availability and low temperature. Is it means that the plants receive Au treatment would perform better than plants in other treatments during the coldest period (Jan to Feb)?

See my detailed comments below:

Detail comments:

Introduction

P.3 L.45. here you mentioned "efficient uptake" (also in the very first sentence of abstract). I don't think you really analyzed nutrient uptake process or efficiency. Maybe you should focus on seasonality.

P.3 L.51. See also Tully 2013 for the driving effect of nutrient demand in plants using internal nutrients. (Tully KL, Wood TE, Schwantes AM, et al. (2013). Soil nutrient availability and reproductive effort drive patterns in nutrient resorption in *pentaclethra macroloba*. Ecology 94:930-940.)

P.3 L.61. Here you mention that no study have examined the effect of seasonally absorbed N in wild plant species. However, your plant materials is obtained from commercial source (P.5 L.102.). Is it a cultivated variety?

P.3 L.63. define "plant performance"

P.3 L.65. There should be a "." after the "var".

P.3 L.65. "Amaryllidacease" should be "Amaryllidaceae".

P.4, L.85-96. The reader expects concisely formulated research questions – or hypotheses here. For example, what kind of function would you expect with the summer/autumn/winter absorbed N, and Why.

Methods

P.7, L.163. no water supply out of the fertilizer supply period?

P.8, L.173. Why do you choose these dates to harvest the plants? Why harvest the C1, Su, Au, SA treatment at 5th January? It seems that the leaf is still growing (Figure 3B.)

Besides, why not harvest the plants in all the treatment at the same time after the leaf senescence? Figure 1 indicates that juvenile plants have leaves from September to next April. After that, the plants shed the leaves, left the storage organ (i.e. bulbs and maybe roots) only. During the leaf senescence, portions of the nutrients will be re-translocated from leaves to storage organs. The energy and nutrient stored in these organs affects the plants future performance. Thus, compare the biomass and the nutrient pool size of the storage organ among treatments at the end of the growing season might be a useful to evaluate the effect of nutrient seasonality on plant performance. I noticed that you would like to examine the effects on leaf "quality", and some of

the measurements might be destructive. However, there are over 20 replicates in each treatment. I think it is possible to test the differences with several replicates.

P.9, L.203. What does the Gleaf stands for? "Leaf amount" in the equation indicates biomass?

P.9, L.203. How do you decide the near-saturating PPFD without measuring the light response curve?

P.13, L.306-313. The Games-howell's method is a non-parametric post-hoc test. Did you perform any homogeneity test? If the data is of equal variance, will parametric test, such as TukeyHSD, achieve a higher statistical power?

P.13, L.306-313. What correlation analysis method was used in figure 5a?

Results:

P.13, L.330. Please consider move the sentences like "The marginal increase in the number of leaves in plants supplied with fertilizer only in autumn (treatment Au) suggests that" to the discussion section.

Discussion

P.16, L. 376. Please define "leaf quality". The term "leaf quality" is obscure, since many parameters, such as N content, LMA, photosynthetic parameters and etc., is significantly different between treatment with nutrient in autumn and other seasons.

P.16, L. 389. Please explain which parameters related with "RuBP- carboxylation and regeneration"

P.17, L395. Remove the trailing "." in "Lycoris."

P.18, L431. "radata" should be "radiata".

Figures

p. 26, Line 611 caption of figure 2. n = 20-22. However, in the methods section, line 177, you mentioned "determined the N concentration in each organ for six plants per treatment." Thus, the replication should be 6? There seem to be some contradictions.

p. 31-35, figures 2-6. Some of the characters in the units of the axis label should be superscripted or subscripted, such as "2" in "cm²" or "CO₂".

Author's Response to Decision Letter for (RSOS-190034.R0)

See Appendix A.

RSOS-190034.R1 (Revision)

Review form: Reviewer 2

Is the manuscript scientifically sound in its present form?

Yes

Are the interpretations and conclusions justified by the results?

Yes

Is the language acceptable?

Yes

Do you have any ethical concerns with this paper?

No

Have you any concerns about statistical analyses in this paper?

No

Recommendation?

Accept with minor revision (please list in comments)

Comments to the Author(s)

The authors have revised the ms. I have no comments other than one suggestion.

L.104-106 I did not find any reasons that lead to your winter hypothesis, and it is a bit conflicting to a previous sentence "...winter leaves are more N-demanding..." (L. 78-79). It will be better if you can add a sentence to explain why you expect that way.

Review form: Reviewer 3

Is the manuscript scientifically sound in its present form?

Yes

Are the interpretations and conclusions justified by the results?

Yes

Is the language acceptable?

No

Do you have any ethical concerns with this paper?

No

Have you any concerns about statistical analyses in this paper?

No

Recommendation?

Reject

Comments to the Author(s)

It's really very hard to read this review. English should be corrected all over the manuscript.

For example: In the abstract itself not clearly written, lengthy sentences with no meaning. Also in one sentence having more "the".

Decision letter (RSOS-190034.R1)

29-Nov-2019

Dear Dr Nishitani:

On behalf of the Editors, I am pleased to inform you that your Manuscript RSOS-190034.R1 entitled "Functional differences in seasonally absorbed nitrogen in a winter-green perennial herb" has been accepted for publication in Royal Society Open Science subject to minor revision in accordance with the referee suggestions. Please find the referees' comments at the end of this email.

The reviewers and Subject Editor have recommended publication, but also suggest some minor revisions to your manuscript. Therefore, I invite you to respond to the comments and revise your manuscript.

- Ethics statement

- Data accessibility

If you wish to submit your supporting data or code to Dryad (<http://datadryad.org/>), or modify your current submission to dryad, please use the following link:
<http://datadryad.org/submit?journalID=RSOS&manu=RSOS-190034.R1>

- Competing interests

- Authors' contributions

- Acknowledgements

- Funding statement

Because the schedule for publication is very tight, it is a condition of publication that you submit the revised version of your manuscript before 08-Dec-2019. Please note that the revision deadline will expire at 00.00am on this date. If you do not think you will be able to meet this date please let me know immediately.

Many thanks & best wishes,

Lianne Parkhouse
Royal Society Open Science
openscience@royalsociety.org

on behalf of the Associate Editor, and Professor Kevin Padian (Subject Editor)
openscience@royalsociety.org

Editor Comments:

Thanks for your resubmission. The reviewers appear to be satisfied with the scientific content but one reviewer finds the language difficult. The best way to approach this is for you to have the manuscript edited by a native speaker of English who is familiar with the subject matter (enough to be able to edit). If the revised version is acceptable linguistically we can proceed to publication. Please note that we will be unable to send the manuscript out for further review so it is important to have it edited before final submission. Thanks very much.

For information about language editing services endorsed by the Royal Society, please follow the link below:
<https://royalsociety.org/journals/authors/language-polishing/>

Reviewer comments to Author:

Reviewer: 2
Comments to the Author(s)

The authors have revised the ms. I have no comments other than one suggestion.

L.104-106 I did not find any reasons that lead to your winter hypothesis, and it is a bit conflicting to a previous sentence "...winter leaves are more N-demanding..." (L. 78-79). It will be better if you can add a sentence to explain why you expect that way.

Reviewer: 3
Comments to the Author(s)

It's really very hard to read this review. English should be corrected all over the manuscript. For example: In the abstract itself not clearly written, lengthy sentences with no meaning. Also in one sentence having more "the".

Author's Response to Decision Letter for (RSOS-190034.R1)

See Appendix B.

Decision letter (RSOS-190034.R2)

20-Dec-2019

Dear Dr Nishitani,

It is a pleasure to accept your manuscript entitled "Functional differences in seasonally absorbed nitrogen in a winter-green perennial herb" in its current form for publication in Royal Society Open Science. The comments of the reviewer(s) who reviewed your manuscript are included at the foot of this letter.

Appendix A

Dear Editors,

Thank you very much for your decision. We have revised the manuscript according to the comments from the reviewer2. The full point-by-point response is supplied below.

Royal Society Open Science - Decision on Manuscript ID RSOS-190034

Associate Editor's comments:

The Editors have made their decision on the basis of their read of the paper and the comments of reviewer 2. As you will see, the second referee is more critical of your paper, but nevertheless sees some merit in further consideration. Please ensure you fully respond to this reviewer's concerns, and incorporate the required changes into your manuscript before you submit. We would also ask that you supply a full point-by-point response to each query/comment of the reviewer and how you've tackled these. Good luck.

Reviewers' Comments to Author:

Reviewer: 2

General comments:

The manuscript is describing that nutrient absorbed during different season was allocated to different organs, and thus affects plant growth in a winter-green perennial herb. The research question is thus in principle of relevance to our understanding the nutrient use strategy of perennial plants. However, there are several issues with the manuscript, which should be addressed:

The introduction could use some more detail and more references to previous works. For example, instead of "... studies have shown different fates of N absorbed at different times ..." (P.3, L.55-56), the reader would be more interested in what kind of differences are there, as well as how and why these differences affect plant's "performance". It would be beneficial if a (or several) specific hypothesis were formulated. This would also make the discussion clearer.

The description of the methods lacks detail in several aspects (please refer to the detailed comments below). I would like to see a more thorough discussion. I think there are several other points worth discussing. For example, besides higher leaf nutrient concentration, the Au treatment leads to significantly higher LMA, which is considered as adaptation to high light availability and low temperature. Is it means that the plants receive Au treatment would perform better than plants in other treatments during the coldest period (Jan to Feb)?

Reply to the general comments

Thank you very much for your careful reading of the manuscript and many helpful comments. We have made revisions according to your comments, as described below.

The introduction section was revised by describing more concretely what kind of differences there are in seasonally absorbed N, and how and why the differences affect plant performance. The sentence you mentioned (P.3, L.55-56 in the original manuscript) was changed to "... studies have shown that allocation and translocation of N within a plant depend on the time of

absorption” (P.3, L.61-62); we avoided using the obscure word “fate”. We wrote clearly that we focused on the roles of N on leaf production (P.4, L.72); we expect that the effects of N on leaf growth and leaf N concentration are different depending on the time of absorption. Why we expect the functional differences and how the differences affect plant performance are explained in the 3rd paragraph (P.4, L.73-90). In the last paragraph we described what we expect specifically with the functions of the seasonally absorbed N in the present study (P.5, L.101-106).

The discussion section was revised by describing more about the role of N absorbed in autumn (P.18 L.439-449). We did not examine the winter growth of the Au plants, however we expect that their higher photosynthetic capacity would lead to a greater growth per leaf area than other treatments in mid-winter. In addition, as you mentioned, higher LMA in Au plants also suggests that they would show better performance (survival and growth) under high light and low temperature conditions in mid-winter. We also included the discussion about the importance of studying the function of N that is translocated from leaves to storage (P.19 L.454-461).

Detail comments:

<Introduction>

1. P.3 L.45. here you mentioned “efficient uptake” (also in the very first sentence of abstract). I don’t think you really analyzed nutrient uptake process or efficiency. Maybe you should focus on seasonality.

Reply to the comment 1

As you pointed out, the expression “efficient uptake” is inappropriate. We changed the phrase “efficient uptake and use of N” to “N uptake in response to its availability and effective N use” (Introduction section P.3 L.49-50) and the beginning of the abstract)

2. P.3 L.51. See also Tully 2013 for the driving effect of nutrient demand in plants using internal nutrients. (Tully KL, Wood TE, Schwantes AM, et al. (2013). Soil nutrient availability and reproductive effort drive patterns in nutrient resorption in pentaclethra macroloba. Ecology 94:930-940.)

Reply to the comment 2

Thank you for your advice. We cited the paper by Tully *et al.* 2013 in the Introduction section (P.3 L.55 and the reference list P.23. L.566-568).

3. P.3 L.61. Here you mention that no study have examined the effect of seasonally absorbed N in wild plant species. However, your plant materials is obtained from commercial source (P.5 L.102.). Is it a cultivated variety?

Reply to the comment 3

***Lycoris radiata* var. *radiata* is a wild plant but is also cultivated as a garden plant. We added the description in the text (P.4 L.92-93).**

4. P.3 L.63. define “plant performance”

Reply to the comment 4

We defined “plant performance” as “plant growth and survival” (P.3 L.69).

5. P.3 L.65. There should be a “.” after the “var”.

6. P.3 L.65. “Amaryllidaceae” should be “Amaryllidaceae”.

Reply to the comments 5 and 6

We corrected them (P4. L71) and also other places with the same mistakes.

7. P.4, L.85-96. The reader expects concisely formulated research questions – or hypotheses here. For example, what kind of function would you expect with the summer/autumn/winter absorbed N, and Why.

Reply to the comments 7

We described what kind of function we expect with the seasonally absorbed N in the last paragraph of the introduction section (P.5 L.101-106). The reason why we expect so is explained in the 3rd paragraph of the Introduction section (P.4 L.73-90).

<Methods>

8. P.7, L.163. no water supply out of the fertilizer supply period?

Reply to the comments 8

All the plants were watered weekly out of the fertilizer supply periods. We added the description in the text. (P.7 L.167-168)

9. P.8, L.173. Why do you choose these dates to harvest the plants? Why harvest the C1, Su, Au, SA treatment at 5th January? It seems that the leaf is still growing (Figure 3B.)

Besides, why not harvest the plants in all the treatment at the same time after the leaf senescence? Figure 1 indicates that juvenile plants have leaves from September to next April. After that, the plants shed the leaves, left the storage organ (i.e. bulbs and maybe roots) only. During the leaf senescence, portions of the nutrients will be re-translocated from leaves to storage organs. The energy and nutrient stored in these organs affects the plants future performance. Thus, compare the biomass and the nutrient pool size of the storage organ among treatments at the end of the growing season might be a useful to evaluate the effect of nutrient seasonality on plant performance. I noticed that you would like to examine the effects on leaf “quality”, and some of the measurements might be destructive. However, there are over 20 replicates in each treatment. I think it is possible to test the differences with several replicates.

Reply to the comments 9

We consider that early January was a good time for harvest in order to evaluate the effect of N absorbed during summer and/or autumn on leaf growth and leaf N concentration and the resulting plant performances (growth and photosynthetic capacity) because (1) leaf growth is almost over by early January and (2) *Lycoris radiata* var. *radiata* shows maximum photosynthesis capacity around early January, in addition, (3) our present study showed the clear difference in N allocation to leaves before and after early January (Fig. 2B).

We chose late march as another sampling time because we wanted to examine the effect of seasonally absorbed N on winter growth before the visible senescence begins.

However, as you mentioned, it is also important to examine the amount and the functions of N which is translocated from leaves to storage in evaluating the effect of nutrient seasonality on plant performance. We pointed the topic as a future research target in the discussion section (P.19 L.454-461).

10. P.9, L.203. What does the G_{leaf} stands for? “Leaf amount” in the equation indicates biomass?

Reply to the comment 10

G_{leaf} stands for the leaf growth (dry mass or area bases) supported by the N that was absorbed during the fertilizer supply period (P.10 L.227). That part of the Materials and methods section explains how to estimate the allocation ratio of seasonally absorbed N in the leaves (Fig. 4A) and the leaf growth supported by unit investment of the N in the leaves (Fig, 4B). However, as the explanations were insufficient and unclear in the original manuscript, we revised the part largely (P.10 L228-234).

11. P.9, L.203. How do you decide the near-saturating PPFD without measuring the light response curve?

Reply to the comment 11

Our preliminary experiments showed that approximately 1200-1500 $\mu\text{mol m}^{-2} \text{s}^{-1}$ PPFD saturated photosynthesis rate of *L. radiata* var. *radiata*. We revised the manuscript by mentioning this preliminary experiment and by describing more detailed procedure for the measurements (P.10 L.240-P.11 L.247). We also corrected our mistake in the original manuscript; the photosynthesis rates were determined at 1500 $\mu\text{mol m}^{-2} \text{s}^{-1}$ PPFD (not 1200-1500 $\mu\text{mol m}^{-2} \text{s}^{-1}$ PPFD) for all the plants (P.11 L.249).

12. P.13, L.306-313. The Games-howell’s method is a non-parametric post-hoc test. Did you perform any homogeneity test? If the data is of equal variance, will parametric test, such as TukeyHSD, achieve a higher statistical power?

Reply to the comment 12

During our preliminary experiments, we encountered some cases where homogeneity of variance was rejected. Because we preferred to use the same statistical method throughout the study, we chose Games-Howell’s method and skipped testing homogeneity of variance. Although Games-Howell’s method has lower statistical power than TukeyHSD when data is of equal variance, we are satisfied with the obtained results, which showed clearly the difference in the properties among fertilizer supply treatments in most of the cases.

13. P.13, L.306-313. What correlation analysis method was used in figure 5a?

Reply to the comment 13

The correlation between leaf area and plant growth in Fig. 5a was detected with Pearson product-moment correlation coefficient. We added this description in the text (P,15 L.351-353)

<Results>

14. P.13, L.330. Please consider move the sentences like “The marginal increase in the number of leaves in plants supplied with fertilizer only in autumn (treatment Au) suggests that” to the discussion section.

Reply to the comment 14

In the revised manuscript, we only wrote the result that the increase in the leaf number was marginal in the treatment Au in the Results section (P.15 L.370 - P.16 L.371); the suggestion from the result is described in the Discussion section (P.20 L.479-482).

<Discussion>

15. P.16, L. 376. Please define “leaf quality”. The term “leaf quality” is obscure, since many parameters, such as N content, LMA, photosynthetic parameters and etc., is significantly different between treatment with nutrient in autumn and other seasons.

Reply to the comment 15

We defined the term “leaf quality” as “leaf N concentration” (P.17 L.413). In the original manuscript we sometimes used the phrase “leaf quantity and quality” however as you pointed out the term quality is obscure. In the revised manuscript we changed the phrase to “leaf growth and leaf N concentration” (e.g., P.2 L. 29-30, P.5 L.99).

16. P.16, L. 389. Please explain which parameters related with “RuBP- carboxylation and regeneration”

Reply to the comments 16

We added the explanation about what parameters relate with RuBP- carboxylation and regeneration in the Materials and methods section (P.11 L.258-260). In addition, we referred Fig. 6J,K (the parameter related with RuBP- carboxylation capacity) and Fig. 6M,N(the parameter related with RuBP- regeneration capacity) in the text in the Discussion section (P.18 L.426-427).

17. P.17, L395. Remove the trailing “.” in “Lycoris.”

18. P.18, L431. “radata” should be “radiata”.

Reply to the comment 17 and 18

We corrected them (P.18 L.436, P.20 L.491) and other places with the same mistakes.

<Figures>

19. p. 26, Line 611 caption of figure 2. $n = 20-22$. However, in the methods section, line 177, you mentioned “determined the N concentration in each organ for six plants per treatment.” Thus, the replication should

be 6? There seem to be some contradictions.

Reply to the comment 19

In Fig. 2 nitrogen content of each organ of each individual plant ($n = 20-22$) was obtained by multiplying the biomass and N concentration of the organ; for the N concentration of each organ we used the mean values for each treatment ($n = 6$).

Because our original manuscript did not describe the method, we revised the relevant part of the Materials and methods section (P.9 L.213-219) and the caption of the Fig. 2 (P.29 L.685-690).

20. p. 31-35, figures 2-6. Some of the characters in the units of the axis label should be superscripted or subscripted, such as “2” in “cm²” or “CO₂”.

Reply to the comment 20

The problem was caused maybe because we uploaded old version of Excel files. We will use other type of files.

Appendix B

Dear Editors,

Thank you very much for your consideration of our manuscript. We have revised the manuscript mostly with the help of an English editing service (Springer Nature Author Services) because major concern about the former version was linguistic problems. The changes made according to the English editing service are shown in red in the revised manuscript (file name: Manuscript YR). Revision was also made according to the suggestion from the reviewer2; we also made minor change in the manuscript where we found mistakes, lack of descriptions etc. in the former version. These changes are highlighted in yellow. The responses to the reviewers are supplied below.

Comment from the reviewer 2

The authors have revised the ms. I have no comments other than one suggestion.

L.104-106 I did not find any reasons that lead to your winter hypothesis, and it is a bit conflicting to a previous sentence "...winter leaves are more N-demanding..." (L. 78-79). It will be better if you can add a sentence to explain why you expect that way.

Reply to the reviewer 2

Thank you very much for your suggestion. We added explanation why we expected that nitrogen (N) absorbed in winter would not be invested in leaves (P.5 L.106-107 in the revised manuscript, file name: Manuscript YR). We expected that seasonally absorbed N would affect leaf production of plants in a little later time, such that N absorbed in summer would affect leaf growth in autumn and N absorbed in autumn would affect leaf N concentration in winter. Likewise we expected that N absorbed in winter would affect leaf production for the next growing season (instead of being invested in the existing winter leaves), and which we considered rational because the environmental temperature increases towards spring.

Comment from the reviewer 3

It's really very hard to read this review. English should be corrected all over the manuscript. For example: In the abstract itself not clearly written, lengthly sentences with no meaning. Also in one sentence having more "the".

Reply to the reviewer 3

We have consulted an English editing service (Springer Nature Author Services). The changes made according to the English editing service are shown in red in the revised manuscript (file

name: Manuscript YR).